# AnyCanvas: Potential Field Guidance for Training-Free Spatial Control in Text-to-Image Diffusion

**Tianyi Xie** [1]  **Zhiyuan Yu** [2]  **Kaihong Huang** [1]  **Beilun Wang** [1]  **Zhaoyang Wang** [1]  **Dian Shen*** [1]

## Abstract

Diffusion-based text-to-image (T2I) models have demonstrated remarkable advancements in generating high-quality images. However, while real-world applications like product packaging and logo design necessitate synthesis within irregular geometries, existing methods struggle to handle such constraints. Therefore, generating complete pictures that conform to arbitrary-shaped canvas constraints while maintaining semantic integrity remains a significant challenge. To address this, we introduce AnyCanvas, a training-free framework that leverages a Mask-to-Potential Field paradigm to convert binary masks into a differentiable potential field, which guides content to naturally converge within target regions. Extensive experiments demonstrate that AnyCanvas achieves 4.23% higher spatial adherence to user-specified constraints while maintaining 99.45% of the semantic fidelity measured by CLIP score, leading to a superior harmonic mean of spatial and semantic metrics. AnyCanvas also exhibits robust generalizability across different model backbones and versatile spatial control objectives.

## 1. Introduction

Diffusion-based text-to-image models (Rombach et al., 2022; Ramesh et al., 2022; Saharia et al., 2022; Karras et al., 2022; Zhang & Tang, 2024; Le et al., 2025) have revolutionized generative AI by demonstrating remarkable capabilities in producing high-quality and diverse images from textual prompts. These models power a wide spectrum of applications, from creative art to professional graphic design (Lee & Chiu, 2023; Tan & Luhrs, 2024), thanks to their ability to synthesize visually appealing content with high semantic alignment (Radford et al., 2021).

Currently, the growing adoption of T2I models in precision-sensitive applications has heightened the need for reliable spatial composition control. This is particularly critical in scenarios such as product packaging, logo design, advertising posters, and short video cover art, where generated content needs to rigorously fit irregular contours or avoid designated foreground objects (Chen et al., 2024b; Ran et al., 2024). However, providing such spatial control remains a significant challenge, as users are often forced to rely on extensive iterative generation and manual filtering (Balasubramanian & Periyaswamy, 2025; Tan & Luhrs, 2024; Oppenlaender, 2022) to approximate the desired output, resulting in workflows that are both inefficient and highly unpredictable.

To this end, existing researches have focused on endowing diffusion models with spatial controllability. For example, *object-level layout conditioning* approaches (Zhang et al., 2023; Mou et al., 2024; Bar-Tal et al., 2023; Mo et al., 2024; He et al., 2023) require explicit manual layouts for spatial control. However, this reliance prevents the model from inferring a plausible internal layout without explicit spatial specifications. Therefore, they fail to produce holistic compositions when only a target canvas shape is provided. In addition, other studies have explored *modifications to attention mechanisms* (Epstein et al., 2023; Liang et al., 2025) to influence the spatial positioning of generated subjects. This demonstrates that semantic content can be spatially modulated by editing cross-attention. However, these methods are restricted to avoiding rectangular regions, failing to generate primary image content within arbitrary and irregular regions. Overall, despite numerous attempts, existing methods still struggle to automatically generate complete paintings within any arbitrary canvas shape.

To address this challenge, we propose AnyCanvas, a unified and training-free framework designed for autonomous image generation within any canvas. Built upon Stable Diffusion, this framework introduces a *Mask-to-Potential Field* paradigm that transforms a user-provided binary mask into a continuously differentiable energy field. During the iterative denoising process, this potential field is dynamically inte-

---

[1]School of Computer Science and Engineering, Southeast University, Nanjing, China [2]State Key Laboratory for Novel Software Technology, Nanjing University, Nanjing, China. Correspondence to: Dian Shen <dshen@seu.edu.cn>.

*Proceedings of the $43^{rd}$ International Conference on Machine Learning*, Seoul, South Korea. PMLR 306, 2026. Copyright 2026 by the author(s).

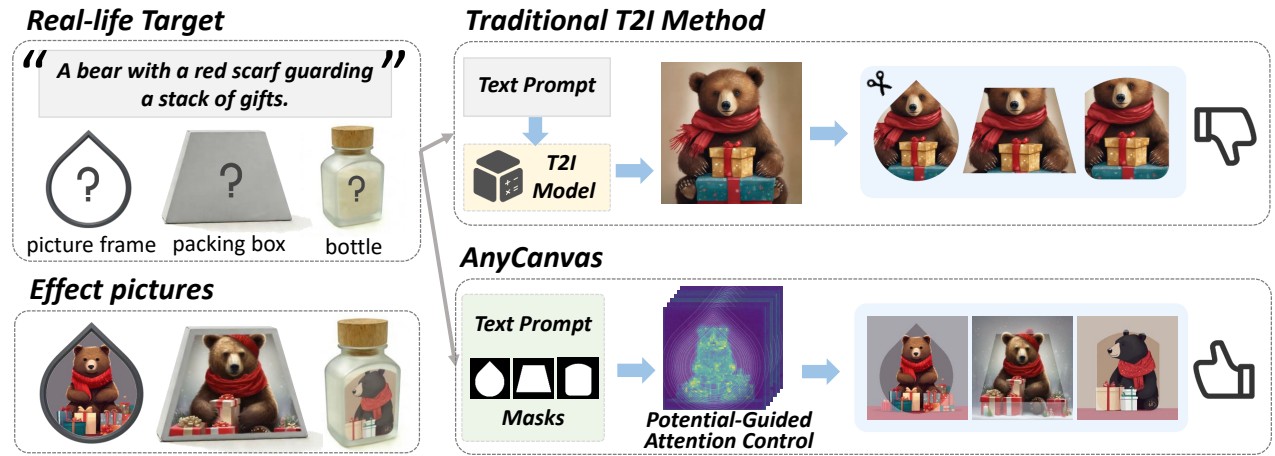

*Figure 1.* Traditional diffusion-based text-to-image methods cannot adapt to canvases of different shapes. Our proposed AnyCanvas addresses this issue through a *Potential-Guided Attention Control* mechanism. Images generated by AnyCanvas can be applied to various real-world scenarios.

grated into the cross-attention layers: the negative gradient of the field guides the direction of the attention distribution, while the magnitude of the potential field modulates the focusing intensity. This physics-inspired system adaptively stabilizes generated content within low-potential regions, ultimately enabling the synthesis of complete artworks that adhere to complex geometric constraints while maintaining semantic integrity.

To validate the effectiveness of our framework, we conduct comprehensive evaluations comparing AnyCanvas against recent leading approaches. Our experimental results demonstrate that AnyCanvas achieves 4.23% higher spatial adherence (Qin et al., 2019) while retaining 99.45% of the semantic fidelity measured by the CLIP score (Radford et al., 2021). Consequently, our method achieves a superior harmonic balance between layout conformity and semantic fidelity, demonstrating the best overall trade-off. Furthermore, the framework exhibits robust generalizability, proving effective across diverse diffusion model backbones and versatile spatial control objectives.

Contributions of this work are summarized as follows:

• We introduce a novel *mask-to-potential field* paradigm that elegantly converts user-provided binary masks into a differentiable potential field, which effectively resolves the conflict between spatial boundaries and semantic coherence of generated content.

• We propose *AnyCanvas*, a training-free framework that dynamically integrates the potential field into the diffusion process. Its plug-and-play nature allows for high-quality and automatic image generation within canvases of arbitrary topological structures.

• We conduct extensive experiments across multiple bench-

marks, demonstrating that *AnyCanvas* achieves superior comprehensive performance on integrated evaluation metrics and exhibits strong generalization capabilities across various diffusion models and spatial control tasks.

## 2. Related Work

**Text-to-Image Generation.** The evolution of Text-to-Image (T2I) generation has progressed from early GAN-based (Reed et al., 2016; Xu et al., 2018) and autoregressive approaches (Ramesh et al., 2021) to the current era dominated by Diffusion Models (Ho et al., 2020; Rombach et al., 2022). With their superior fidelity and diversity, diffusion models have become the cornerstone of generative AI. Recently, the research focus has shifted towards Controllable Generation (Cao et al., 2025; Zhang et al., 2023; Ma et al., 2024), aiming for precise intervention over style (Chung et al., 2024; Wang et al., 2025), subject (Ma et al., 2024), and layout. Our work specifically targets the challenge of flexible spatial control within this paradigm.

**Spatial Controllable Generation.** Learning-based methods inject spatial priors by training auxiliary modules (Zhang et al., 2023; Mou et al., 2024; Li et al., 2024b; Wang et al., 2024) or fine-tuning attention layers (Li et al., 2023; Cheng et al., 2024). Recent trends also explore leveraging Multi-modal LLMs for layout planning (Fang et al., 2025; Yang et al., 2024; Mi et al., 2025). While effective on predefined tasks, these approaches often struggle to generalize to open-vocabulary shapes (Chen et al., 2024c; Ran et al., 2024) and require costly retraining when transferring to different base models (Li et al., 2024a; Wang et al., 2025).

Training-free methods intervene directly during inference to bypass training costs. This category includes optimization-

based guidance (Xie et al., 2023; Bar-Tal et al., 2023; Mo et al., 2024; Ohanyan et al., 2024; Xiao et al., 2023) (updating latents via layout losses) and attention-based modulation (Hertz et al., 2023; Liang et al., 2025; Phung et al., 2024; Cao et al., 2023). However, these existing approaches face a fundamental limitation: they rely on either discrete constraints or discontinuous heuristics, limiting their ability to conform to arbitrary and complex geometries while maintaining semantic integrity (Epstein et al., 2023; He et al., 2023; Chen et al., 2024b), highlighting the need for constructing a principled, continuous control framework.

# 3. Preliminary

## 3.1. Diffusion Models

Diffusion models are a class of generative models that learn to generate data by iteratively denoising a random noise variable. The process consists of a forward noising process $q$ and a reverse denoising process $p_\theta$.

**Forward Process.** Given a data sample $\mathbf{x}_0 \sim q(\mathbf{x}_0)$, the forward process produces a sequence of increasingly noisy latent variables $\mathbf{x}_1, \ldots, \mathbf{x}_T$ by adding Gaussian noise at each timestep $t$ according to a variance schedule $\beta_t \in (0, 1)$:

$$q(\mathbf{x}_t | \mathbf{x}_{t-1}) = \mathcal{N}(\mathbf{x}_t; \sqrt{1 - \beta_t}\mathbf{x}_{t-1}, \beta_t\mathbf{I}). \quad (1)$$

**Reverse Process.** Generation starts from pure noise $\mathbf{x}_T \sim \mathcal{N}(\mathbf{0}, \mathbf{I})$ and is defined by a learned Markov chain that progressively removes noise:

$$p_\theta(\mathbf{x}_{t-1} | \mathbf{x}_t) = \mathcal{N}(\mathbf{x}_{t-1}; \mu_\theta(\mathbf{x}_t, t), \Sigma_\theta(\mathbf{x}_t, t)), \quad (2)$$

where the model $\theta$ predicts the parameters of the Gaussian transition. A common parameterization is to predict the additive noise $\epsilon_\theta(\mathbf{x}_t, t)$ for the given $\mathbf{x}_t$. The training objective simplifies to a mean-squared error loss:

$$\mathcal{L}_{\text{simple}} = \mathbb{E}_{t, \mathbf{x}_0, \epsilon \sim \mathcal{N}(0, \mathbf{I})} \left[ \|\epsilon - \epsilon_\theta(\mathbf{x}_t, t)\|^2 \right]. \quad (3)$$

Our work leverages the internal cross-attention mechanisms of pre-trained latent diffusion models to achieve spatial control without altering this core denoising process.

## 3.2. Text-to-Image Generation

Text-to-Image (T2I) generation synthesizes visual content conditioned on natural language descriptions. In diffusion models, a text prompt $\mathcal{P}$ is encoded into a continuous vector $\mathbf{c} = \tau(\mathcal{P})$, which guides the generative process via the conditional denoising objective:

$$\mathcal{L}_{\text{cond}} = \mathbb{E}_{t, \mathbf{x}_0, \mathbf{c}, \epsilon} \left[ \|\epsilon - \epsilon_\theta(\mathbf{x}_t, t, \mathbf{c})\|^2 \right]. \quad (4)$$

This conditioning enables high-level semantic alignment between the generated image and the prompt.

**Cross-Attention Mechanism.** The key mechanism for semantic grounding is the cross-attention layer. Given a visual feature map $\mathbf{z} \in \mathbb{R}^{H_f \times W_f \times C}$ and text embeddings $\mathbf{c} \in \mathbb{R}^{L \times d}$, the spatial attention map $A$ is computed as:

$$A = \text{softmax}\left( \frac{\mathbf{z}\mathbf{W}_Q(\mathbf{c}\mathbf{W}_K)^\top}{\sqrt{d_k}} \right), \quad (5)$$

where $\mathbf{W}_Q, \mathbf{W}_K$ are projection matrices. $A \in \mathbb{R}^{H_f \times W_f \times L}$ dictates where textual concepts are manifested in the image, providing a latent spatial prior for object placement.

## 3.3. Spatial Control for T2I

This work addresses the problem of imposing arbitrary spatial constraints on pre-trained text-to-image diffusion models. Given a text prompt $\mathcal{P}$ and a binary mask $M \in \{0, 1\}^{H \times W}$ defining a target region, our goal is to generate an image $I$ where the semantic content aligned with $\mathcal{P}$ is precisely confined within $M$. Let $S(I) \in [0, 1]^{H \times W}$ represent a semantic saliency map derivable from cross-attention activations that captures the spatial distribution of prompt-relevant content. The objective is to maximize the concentration of salient features inside the mask while preserving semantic fidelity:

$$\max \ \mathcal{J}(I) = \frac{\sum_{(x,y)} S_{x,y}(I) \cdot M_{x,y}}{\sum_{(x,y)} S_{x,y}(I)} \quad \text{s.t.} \quad \mathcal{F}(I, \mathcal{P}) \geq \tau, \quad (6)$$

where $\mathcal{F}(I, \mathcal{P})$ quantifies semantic alignment (e.g., CLIP score) and $\tau$ is a quality threshold. This formulation highlights the core challenge: achieving precise spatial control without compromising semantic integrity.

# 4. Method

To achieve precise spatial control, we propose AnyCanvas, a unified framework designed for T2I generation within arbitrary canvases. In AnyCanvas, we dynamically modify the cross-attention maps through an affine transformation, which consists of a translation and a scaling operation derived from the potential field. This transformation is applied at each denoising step to align the semantic content with the target region defined by the mask. Specifically, for each cross-attention layer, we compute an affine transformation matrix $T$ based on a displacement vector $\mathbf{\Delta p}$ and a scaling factor $s$, with the centroid $\mathbf{p_c}$ of the attention map as the reference point. The transformation is then applied to the attention map $A_t$ to produce the modified map $A'_t$, with:

$$\mathcal{T} \leftarrow \mathcal{B}(\Delta\mathbf{p}, \mathbf{p}_c, s) \quad (7)$$
$$A'_t \leftarrow \mathcal{A}(A_t, \mathcal{T}) \quad (8)$$

where $\mathcal{B}$ denotes the function that builds an affine transformation matrix incorporating translation by $\Delta\mathbf{p}$ and scaling

## Potential Field Construction

## Dynamic Guidance

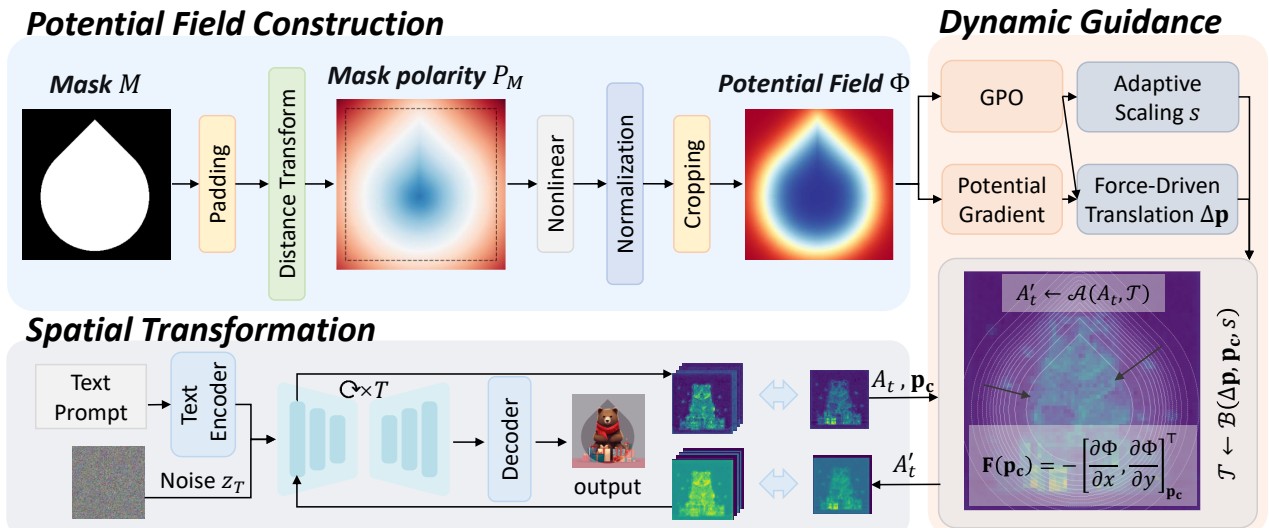

*Figure 2.* Overall Framework of AnyCanvas.

by $s$ around the centroid $\mathbf{p}_c$, and $\mathcal{A}$ denotes the function that applies this transformation to the attention map. This process adaptively steers the attention distribution toward the low-potential region of the mask, forming a closed-loop control system. Overall framework is illustrated in Figure 2.

### 4.1. Construction of the Mask Potential Field

To compute the displacement vector $\Delta\mathbf{p}$ and the scaling factor $s$ required for the affine transformation, we first construct a continuous and differentiable potential field from the discrete binary mask. This field provides a geometric prior for spatial guidance. Our construction is based on the Signed Distance Function (SDF). Specifically, for a binary mask $M$, the SDF $d_M(x, y)$ is defined as the shortest Euclidean distance from pixel $(x, y)$ to the mask boundary, with the convention that $d_M > 0$ inside the mask, $d_M < 0$ outside, and $d_M = 0$ on the boundary. To construct the potential field, we first define the mask polarity:

$$P_M(x, y) \triangleq \operatorname{sgn}(M(x, y) - 0.5) \tag{9}$$

The energy at each position is then given by

$$E(x, y) = P_M(x, y) \cdot f\left(\frac{|d_M(x, y)|}{\sigma}\right) \tag{10}$$

where $\operatorname{sgn}(\cdot)$ is the sign function, $f : \mathbb{R}^+ \to [0, 1]$ is a monotonically decreasing nonlinear function satisfying $f(0) = 1$ and $\lim_{z\to\infty} f(z) = 0$, and $\sigma > 0$ is a smoothing factor controlling the decay rate of the potential. The potential field $\Phi$ is normalized to the range $[-1, +1]$ for consistent

gradient magnitudes across arbitrary mask shapes:

$$\Phi(x, y) = 2 \cdot \frac{E(x, y) - \min_{(x,y)} E(x, y)}{\max_{(x,y)} E(x, y) - \min_{(x,y)} E(x, y)} - 1 \tag{11}$$

Physically, the sign term ensures that $\Phi$ is negative inside the mask (i.e., $M = 1$), forming an attractive potential well for content, while outside ($M = 0$), $\Phi$ is positive, creating a repulsive potential barrier. The nonlinear term $f$ guarantees continuity and smoothness at the mask boundary ($d_M = 0$), with $\Phi = 0$ at the boundary and asymptotically approaching $\pm 1$ as the distance increases. The parameter $\sigma$ controls the smoothness of the potential transition. This potential field establishes a global minimum within the target region, forming the physical basis for guiding the spatial distribution of generated content.

**Robust Solution for Image Boundary Issues:** In practical experiments, we find that if the mask $M$ does not extend to the image boundary, the potential at edge pixels computed by Equation (10) may be lower than inside the mask, creating a spurious global minimum that misguides generation. To address this, we pre-process the mask by padding it with a band of width $w_{\text{pad}} = 10$ pixels on all sides, marking the padded area as prohibited ($M = 0$), yielding $M_{\text{padded}}$. The SDF and potential field $\Phi$ are then derived from $M_{\text{padded}}$. This operation ensures the user-defined mask interior ($M = 1$) is the sole potential well, thus preventing content overflow simply and effectively without additional losses or constraints.

### 4.2. Dynamic Guidance via the Potential Field

From a physical perspective, the proposed guidance mechanism can be interpreted as a potential-driven control process,

where the spatial evolution of semantic attention follows the descent of an underlying potential field. The potential field $\Phi$ provides a continuous geometric prior for spatial guidance, which is dynamically injected into the diffusion model during the iterative denoising process. This integration enables adaptive control over the generation by influencing the attention mechanisms. To ensure robust spatial guidance, these core components are inspired by physical principles rather than purely empirical heuristics.

**Spatial Centroid $\mathbf{p}_c$:** The spatial attention map $A \in \mathbb{R}^{H_f \times W_f}$ generated by the cross-attention layer reflects the expected distribution of semantic concepts from text prompt, with high-activation regions denoting probable object locations. The spatial centroid $\mathbf{p}_c = (x_c, y_c)$ is computed as the first-order moment of the attention map. Specifically, for a discrete attention map $A_t$ with spatial dimensions $H_f \times W_f$, the centroid coordinates are given by:

$$x_c = \frac{\sum_{i=1}^{H_f} \sum_{j=1}^{W_f} j \cdot A_{ij}}{\sum_{i=1}^{H_f} \sum_{j=1}^{W_f} A_{ij}}, \ y_c = \frac{\sum_{i=1}^{H_f} \sum_{j=1}^{W_f} i \cdot A_{ij}}{\sum_{i=1}^{H_f} \sum_{j=1}^{W_f} A_{ij}}, \tag{12}$$

Dynamic modulation of $\mathbf{p}_c$ simulates the motion of a physical particle in a potential, enabling precise indirect control over the spatial layout of generated content.

**Displacement $\Delta\mathbf{p}$:** According to classical physics, the force $\mathbf{F}$ on a particle located at $\mathbf{p}_c$ in the potential field $\Phi$ is the negative gradient of the potential at that point, pointing in the direction of steepest potential descent:

$$\mathbf{F}(\mathbf{p}_c) = -\nabla\Phi(\mathbf{p}_c) = -\left[\frac{\partial\Phi}{\partial x}, \frac{\partial\Phi}{\partial y}\right]^{\top}_{\mathbf{p}_c} \tag{13}$$

However, in practice, directly using the force magnitude for displacement can cause instability. In flat or near-boundary regions of the potential field, the gradient magnitude can become very small or change abruptly, leading to inefficient centroid movement. To address this, we introduce the *Global Potential Offset* to decouple the displacement magnitude from the raw gradient.

In global potential offset, the target potential is defined as the minimum value within the mask region $\Omega_{\mathrm{mask}}$, serving as a reference point for attraction:

$$\Phi_{\mathrm{tgt}} = \min_{(x,y)\in\Omega_{\mathrm{mask}}} \Phi(x, y) \tag{14}$$

The potential offset $\Delta\Phi$ at the centroid position is then computed as the difference between the current potential $\Phi(\mathbf{p}_c)$ and target potential $\Phi_{\mathrm{tgt}}$ to ensure non-negativity:

$$\Delta\Phi = \Phi(\mathbf{p}_c) - \Phi_{\mathrm{tgt}} \geq 0 \tag{15}$$

The displacement magnitude $m$ is formulated as a monotonically increasing nonlinear function of potential offset $\Delta\Phi$ to promote smooth and stable motion. For practical implementation, it is computed through following:

$$m = \kappa \cdot (\exp(\Delta\Phi) - 1) \tag{16}$$

where $\kappa$ is a global coefficient that modulates the overall movement strength. This design ensures zero displacement when $\Delta\Phi = 0$, and $m$ increases smoothly as $\Delta\Phi$ grows, thereby enabling stable and efficient convergence.

Finally, the displacement vector $\Delta\mathbf{p}$ is derived by combining the direction unit vector $\hat{\mathbf{d}}$ with the magnitude $m$, resulting in a controlled and adaptive update:

$$\Delta\mathbf{p} = m \cdot \hat{\mathbf{d}} \quad \text{with} \ \hat{\mathbf{d}} = \frac{\mathbf{F}}{\|\mathbf{F}\|_2 + \epsilon} \tag{17}$$

where $\epsilon > 0$ is applied to prevent division by zero.

**Scaling $s$:** While displacement operation can effectively reposition the attention centroid, it may not sufficiently adapt the spatial coverage of the attention map to irregular mask contours. To ensure a harmonious fit between the semantic content and the target region, we introduce an adaptive scaling mechanism that dynamically adjusts the map's extent based on local potential conditions.

Following the translation step, the predicted centroid position is given by $\mathbf{p}' = \mathbf{p}_c + \Delta\mathbf{p}$. We then evaluate the potential $\Phi(\mathbf{p}')$ at this new location as follows:

$$\Delta\Phi_{\mathrm{scale}} = \Phi(\mathbf{p}') - \Phi_{\mathrm{tgt}} \tag{18}$$

This potential difference serves as the basis for dynamically determining the scaling factor $s$ with:

$$s = \mathrm{clamp}\left(\exp(-\lambda \cdot \Delta\Phi_{\mathrm{scale}}), \ S_{\mathrm{min}}, \ 1.0\right). \tag{19}$$

Here, $\lambda > 0$ is a hyperparameter that controls the sensitivity of the scaling response, and $S_{\mathrm{min}}$ is a preset lower bound (set as 0.5 for implementation) to prevent excessive contraction. This ensures that attention map maintains its original scale when the predicted centroid lies in a low-potential region, and is contracted inward to conform to the mask shape as the centroid approaches high-potential regions.

### 4.3. Spatial Transformation of Attention Maps

To implement the geometric guidance derived from the potential field, we implement the control mechanism through an affine-based spatial transformation. The process is formally defined by two core functions: the transformation matrix constructor $\mathcal{B}$ and the spatial applicator $\mathcal{A}$.

The constructor function $\mathcal{B}(\Delta\mathbf{p}, \mathbf{p}_c, s)$ synthesizes an affine transformation matrix $\mathcal{T}$ that integrates translation by $\Delta\mathbf{p}$ and isotropic scaling by $s$ around the centroid $\mathbf{p}_c = (c_x, c_y)$. The matrix is explicitly constructed as:

$$\mathcal{T} = \begin{bmatrix} \frac{1}{s} & 0 & (1-\frac{1}{s})c_x - \frac{1}{s}\Delta x \\ 0 & \frac{1}{s} & (1-\frac{1}{s})c_y - \frac{1}{s}\Delta y \end{bmatrix} \tag{20}$$

Here, the isotropic scaling is centered at the centroid $\mathbf{p}_c$ to adapt the spatial extent of the attention map, while the displacement component $\Delta\mathbf{p}$ shifts the map toward the low-potential region identified by the mask.

The applicator function $\mathcal{A}(A_t, \mathcal{T})$ then warps the attention map $A_t$ using the predefined affine matrix $\mathcal{T}$. For any spatial coordinate $\mathbf{u}$, the transformed attention map is computed via inverse coordinate mapping:

$$A'_t(\mathbf{u}) = A_t(\mathcal{T}^{-1}\mathbf{u}) \tag{21}$$

This operation redistributes the attention activations spatially to conform to the target region, effectively translating the potential field's geometric constraints into precise adjustments of the semantic layout.

**Selective Attention Modulation:** To reduce computational overhead and better preserve semantic fidelity, we only applies affine transformations to tokens that are significantly misaligned with the target mask. For token $t$'s attention map $A_t$, we define a conflict ratio as:

$$r = \frac{\sum_{(x,y)\notin\Omega_{\text{mask}}} A_t(x,y)}{\sum_{(x,y)} A_t(x,y)}, \tag{22}$$

where $\Omega_{\text{mask}}$ is the mask region. We select $K$ tokens with $r > \delta$ and apply affine transformations using token-specific coefficients. This targeted modulation corrects their spatial distribution towards the mask's low-potential area while leaving other tokens unaffected.

### 4.4. Computational Complexity Analysis

The single-step denoising process of Stable Diffusion contains $L$ attention layers. For a cross-attention layer with spatial resolution $H \times W$, hidden dimension $D$, and text length $N$, the per-layer complexity is:

$$\mathcal{O}_{\text{Layer}} = \mathcal{O}(HW \cdot D^2) + \mathcal{O}(HW \cdot N \cdot D) \approx \mathcal{O}(HW \cdot D^2), \tag{23}$$

Since $D \gg N$, the quadratic term dominates. In Any-Canvas, the potential field is built once before inference with complexity $O(HW \log HW)$. During inference, each cross-attention layer performs potential-guided operations on $K$ selective tokens, the additional per-layer complexity is thus approximated as $\mathcal{O}_{\text{ours}} \approx \mathcal{O}(C \cdot K \cdot HW)$. Compared to the baseline, the overhead satisfies

$$\frac{\mathcal{O}_{\text{ours}}}{\mathcal{O}_{\text{Layer}}} \propto \frac{K}{D^2}, \quad K \ll D^2, \tag{24}$$

This indicates that the added computation from the potential-guided mechanism is asymptotically negligible.

## 5. Experiment

### 5.1. Experimental Setup

**Implementation:** All experiments use two NVIDIA RTX A6000 GPUs (96GB total). Our training-free method is implemented on top of Stable Diffusion using the `diffusers` library without any fine-tuning. We generate images at $1024 \times 1024$ resolution with 50 DDIM steps. The hyperparameter sensitivity analyses and the corresponding hyperparameter settings are included in the Appendix.

**Datasets:** We select three benchmarks for evaluation. Prompt-to-Prompt Template (PTP) (Hertz et al., 2023; Liang et al., 2025) assesses the model's response to spatial geometric constraints; DiffusionDB (Liang et al., 2025) evaluates generalization in real-world scenarios, covering diverse styles to test open-domain performance; PartiPrompts (P2) (Yu et al., 2022) measures overall generative capability across multiple categories and difficulty levels. In addition, a Prompt-Mask Decoupling strategy is applied, where we assign each prompt to multiple masks with varied topologies to rigorously test adaptability to arbitrary geometric shapes.

**Baselines:** To evaluate the efficacy of AnyCanvas in arbitrary canvas generation, we compared it against representative baselines, including Stable Diffusion XL (SDXL) (Rombach et al., 2022), TextCenGen (Liang et al., 2025), FLUX.1 (Labs et al., 2025), and MultiDiffusion (Bar-Tal et al., 2023). Since the latter three methods are not inherently designed for irregular spatial control, we applied targeted adaptations to ensure a fair comparison. More implementation details and specific settings can be found in the Appendix.

**Metrics:** Generation quality is evaluated using three metrics. CLIP Score (Radford et al., 2021) measures semantic alignment between the generated image and the text prompt. Canvas-IoU quantifies spatial adherence between the salient region $S$ extracted from the generated image using BAS-Net (Qin et al., 2019), a predict-refine architecture optimized via a hybrid loss, and the target mask $M$:

$$\text{Canvas-IoU} = \frac{|S \cap M|}{|S \cup M|}. \tag{25}$$

The Image-Canvas Quality Score (ICQS) integrates semantic fidelity and spatial accuracy by taking the harmonic mean of CLIP Score and Canvas-IoU:

$$\text{ICQS} = \frac{2 \cdot \text{CLIP} \cdot \text{Canvas-IoU}}{\text{CLIP} + \text{Canvas-IoU}}. \tag{26}$$

All metrics are positively correlated with generation quality, where higher values indicate better performance.

*Table 1.* Comparison of AnyCanvas against different baselines.

| Method | PTP Dataset | | | DDB Dataset | | | P2 Dataset | | |
|---|---|---|---|---|---|---|---|---|---|
| | C-IoU | Clip Score | ICQS | C-IoU | Clip Score | ICQS | C-IoU | Clip Score | ICQS |
| SDXL | 34.48 | 32.75 | 31.31 | 41.94 | 32.39 | 34.48 | 37.80 | 32.18 | 32.02 |
| TextCenGen | 35.63 | 32.60 | 31.71 | 40.62 | 32.78 | 33.90 | 36.74 | 32.31 | 31.44 |
| Flux.1 | 37.34 | 31.82 | 31.62 | 45.52 | 29.28 | 33.51 | 41.32 | 31.50 | 33.23 |
| MultiDiffusion | 35.55 | 30.87 | 29.54 | 40.59 | 30.93 | 32.13 | 36.96 | 31.18 | 30.44 |
| AnyCanvas | 36.44 | 32.55 | **32.68** | 43.17 | 32.07 | **35.67** | 39.34 | 32.16 | **33.49** |

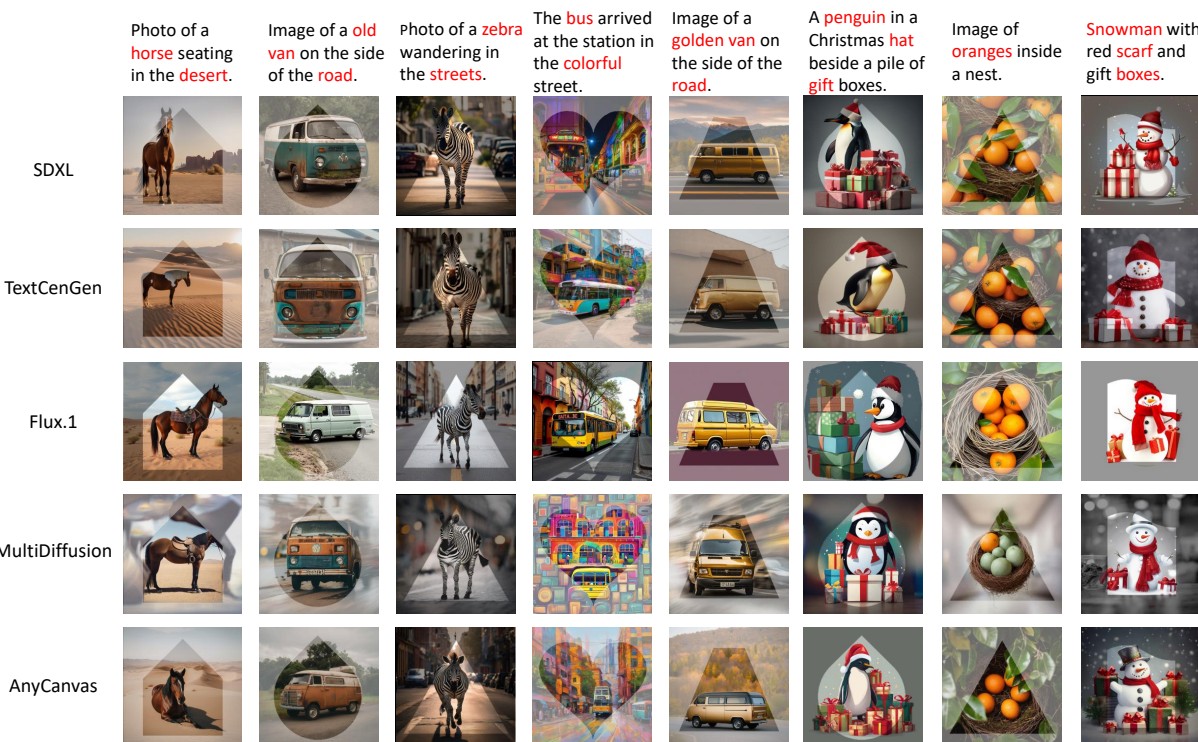

*Figure 3.* Case presentation of different T2I methods across diverse canvas masks.

## 5.2. Result Analysis

As shown in Table 1, AnyCanvas achieves a more balanced and stable trade-off between spatial control and semantic consistency. Compared with the SDXL baseline, AnyCanvas attains substantial improvements in the C-IoU metric, which measures spatial positional control, achieving relative gains of 5.68%, 2.93%, and 4.07% on the PTP, DDB, and P2 datasets, respectively, while retaining an average of 99.45% of the CLIP score achieved by the base model across datasets. By jointly preserving geometric precision and semantic quality, AnyCanvas consistently outperforms existing methods on the comprehensive ICQS metric. In contrast, although Flux.1 achieves relatively high C-IoU scores on datasets such as DDB, these gains are typically accompanied by a notable degradation in CLIP Score, resulting in no advantage on the ICQS metric. The Appendix presents a detailed trade-off analysis of the metrics and supplementary comparisons with additional spatial control methods, demonstrating the superior balance achieved by AnyCanvas.

We present some of text-to-image experiment results in Figure 3. As illustrated, SDXL exposes the fundamental limitation of generating without explicit spatial guidance. TextCenGen, since being designed for rectangular bounding boxes, is unsuitable for canvases with complex shapes. In FLUX.1, we use linguistic prompts to indicate mask positions, but diffusion models fail to understand it and exhibit poor spatial adherence. MultiDiffusion is characterized by stitching artifacts and a cartoon-like appearance. This indicates that existing methods have various problems when dealing with this scenario. In comparison, our proposed AnyCanvas can effectively solve these problems.

## 5.3. Generalization Analysis

To validate the generalizability of AnyCanvas across different diffusion model architectures, we conducted experiments on various Stable Diffusion (SD) versions. The results in Table 2 demonstrate that our framework functions as a versatile plug-and-play module, consistently improving the overall generation quality across all tested models. Specifically, AnyCanvas achieves consistent growth in C-IoU metrics across all baselines, demonstrating the strong robustness of our spatial constraint mechanism across different architectures. Regarding the comprehensive metric ICQS, while the improvements on earlier models are modest, the performance gains become more significant on advanced models. For instance, AnyCanvas achieves relative improvements in ICQS of 4.70% on SD2.0 and 4.38% on SDXL. Furthermore, to demonstrate that our method is not limited to U-Net architectures, we also evaluated it on a mainstream Diffusion Transformer (DiT) model, PixArt-$\alpha$ (Chen et al., 2024a), where it achieved a 4.14% relative improvement in ICQS. These consistent improvements provide empirical evidence for the generalizability of our framework. Furthermore, extending this potential-guided control mechanism to the joint attention layers of modern MMDiT architectures also presents certain theoretical feasibility for future exploration (Park et al., 2025) .

Table 2. Generalization analysis results.

| Method | C-IoU | Clip | ICQS | Δ (%) |
|---|---|---|---|---|
| SD1.4 | 27.50 | 31.36 | 26.74 | - |
| AnyCanvas | 29.42 | 30.66 | 26.88 | ↑ 0.52 |
| SD1.5 | 26.62 | 31.57 | 26.00 | - |
| AnyCanvas | 27.89 | 30.60 | 26.07 | ↑ 0.27 |
| SD2.0 | 26.04 | 31.66 | 24.87 | - |
| AnyCanvas | 27.90 | 30.41 | 26.04 | ↑ 4.70 |
| SD2.1 | 29.30 | 31.47 | 26.72 | - |
| AnyCanvas | 29.74 | 30.28 | 27.48 | ↑ 2.84 |
| SDXL | 34.48 | 32.75 | 31.31 | - |
| AnyCanvas | 36.44 | 32.55 | 32.68 | ↑ 4.38 |
| PixArt-$\alpha$ | 35.06 | 32.77 | 31.39 | - |
| AnyCanvas | 36.00 | 31.93 | 32.69 | ↑ 4.14 |

## 5.4. Ablation Study

To rigorously evaluate the contribution of each component in our framework, we conduct an ablation study, with results summarized in Table 3. As demonstrated, the removal of either the translation module (*w/o trans*) or the scaling module (*w/o scaling*) leads to a notable degradation in spatial adherence, while the impact on semantic fidelity is relatively small. This indicates that while each module is critical for geometric precision, while the model's ability to understand textual semantics is robust to architectural changes. Moreover, only when both modules are employed can the ICQS metric achieve a substantial improvement over the baseline.

These results validate the necessity of our designed modules.

Table 3. Ablation study of different components

| Method | C-IoU | Clip Score | ICQS |
|---|---|---|---|
| w/o all | 34.48 | 32.75 | 31.31 |
| w/o trans | 35.09 | 32.56 | 31.38 |
| w/o scaling | 35.78 | 32.57 | 31.84 |
| ours | 36.44 | 32.55 | 32.68 |

## 5.5. Task Versatility Analysis

Previous experiments primarily validated the capability of AnyCanvas to fit content to specific contours. To demonstrate that AnyCanvas supports diverse spatial control objectives, we address a scenario where real-world applications often present a complementary requirement: generating an image while effectively suppressing content generation within predefined regions. AnyCanvas naturally supports this through a unified mask definition, requiring no architectural modifications or explicit inversion steps. Users simply need to provide an appropriate mask, and the resulting potential field forms a high-potential barrier in these regions to repel semantic content.

To evaluate this capability, we conducted quantitative experiments using diverse mask layouts designed for content suppression. AnyCanvas remains effective in this challenging setting. As shown in Table 4, AnyCanvas achieves a relative improvement of 9.65% in C-IoU compared to the baseline, indicating improved spatial avoidance accuracy. Meanwhile, it retains 98.63% of the original CLIP Score, suggesting that semantic quality is largely preserved. Consequently, AnyCanvas attains a 4.39% relative improvement in the comprehensive ICQS metric. Furthermore, Figure 4 visually compares the SDXL baseline with and without AnyCanvas, illustrating that the proposed method consis-

Table 4. Results of content suppression.

| Method | C-IoU | Clip | ICQS |
|---|---|---|---|
| SDXL | 20.52 | 32.90 | 24.15 |
| AnyCanvas | 22.50 | 32.45 | 25.21 |

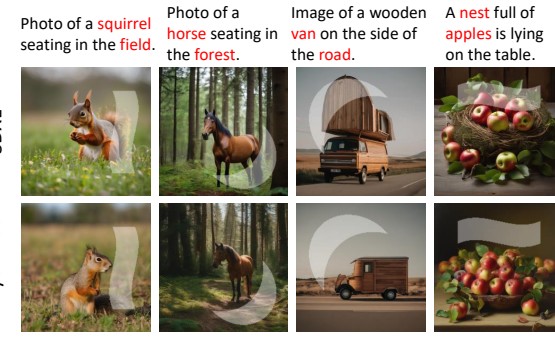

Figure 4. Case presentation of content suppression tasks.

tently supports content suppression under arbitrary-shaped regions. These results provide empirical evidence for the task versatility of AnyCanvas across different spatial control objectives.

## 6. Conclusion

In this paper, we introduce AnyCanvas, a novel training-free framework that addresses the critical challenge of achieving precise spatial control in text-to-image generation within arbitrarily shaped canvases. Motivated by the limitations of existing methods that often enforce discrete constraints or require costly fine-tuning, we proposed a physics-inspired Mask-to-Potential Field paradigm. This core innovation elegantly transforms a binary mask into a differentiable potential field, whose gradient dynamically guides the diffusion process through principled affine transformations of the cross-attention maps. Extensive experiments demonstrate that AnyCanvas achieves superior performance across diverse benchmarks and diffusion model architectures.

## Acknowledgements

This work was supported by Guangxi Science and Technology Achievement Transformation Program No.ZG2504240019, Science and Technology Major Special Program of Jiangsu Grants No. BG2024028, the Natural Science Foundation of Jiangsu Province under Grant No. BK20253020, BK20230083, National Natural Science Foundation of China under Grant No.62522205, 62272101, and in part by the Collaborative Innovation Center of Novel Software Technology and Industrialization, the Big Data Computing Center of Southeast University.

## Impact Statement

This paper presents work whose goal is to advance the field of Machine Learning. There are many potential societal consequences of our work, none which we feel must be specifically highlighted here.

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

# A. Implementation Details

In our experiments, we generated a total of 26,775 synthesized images for evaluation, comprising 3,096 on PTP, 8,991 on DDB, and 14,688 on P2. Since most existing T2I spatial control methods are not natively designed for arbitrary canvas constraints, we introduce specific adaptations to apply them to this task. The detailed configurations for each baseline are as follows:

**Stable Diffusion XL (SDXL) (Rombach et al., 2022).**    We utilize the official implementation of SDXL-Base-1.0 as the reference model. It takes the text prompt as input without any explicit spatial masks, serving as a baseline to assess the native generation quality and the difficulty of the prompts.

**TextCenGen (Liang et al., 2025).**    TextCenGen relies on a force-directed movement module to optimize object layout, but this mechanism is inherently constrained to rectangular bounding boxes and cannot handle non-convex or irregular shapes. To adapt it for our task, we disable the movement module and retain only its spatial excluding constraint. This allows us to evaluate the model's ability to restrict cross-attention activations within the fixed, user-provided arbitrary masks.

**FLUX.1 (Peebles & Xie, 2023; Liu & Gong, 2023; Labs, 2024; Labs et al., 2025).**    As a state-of-the-art T2I model with strong semantic understanding, FLUX.1 represents the upper bound of prompt-following capabilities. Since it does not support mask inputs directly, we employ prompt engineering to simulate spatial control. Specifically, we append explicit spatial instructions to the user prompt in the format: "*Place the main subject within a {shape} shape*" . This evaluates whether the model can spontaneously align content with geometric constraints purely through semantic instructions.

**MultiDiffusion (Bar-Tal et al., 2023).**    MultiDiffusion enables region-based generation through a panoramic denoising process. We adapt it by defining the target arbitrary mask as the foreground region, which is controlled by the specific subject prompt. To handle the irregular boundaries, we do not provide a specific semantic prompt for the background (outside the mask). This setup tests the method's flexibility in filling irregular regions while maintaining global coherence.

# B. Metric for Trade-off Analysis

As visualized in Figure 5, AnyCanvas achieves the most robust trade-off between spatial adherence and semantic fidelity, consistently pushing the optimal iso-utility frontier.

To quantify the balance between spatial controllability and generation quality, we adopt an *iso-utility analysis* based on the geometric mean, a standard approach in multi-objective optimization.

**Metric Formulation.** Let $x$ denote the spatial accuracy (C-IoU) and $y$ denote the semantic fidelity (CLIP Score). To ensure commensurability between metrics with different scales, we first normalize them to the unit interval $[0, 1]$ based on the

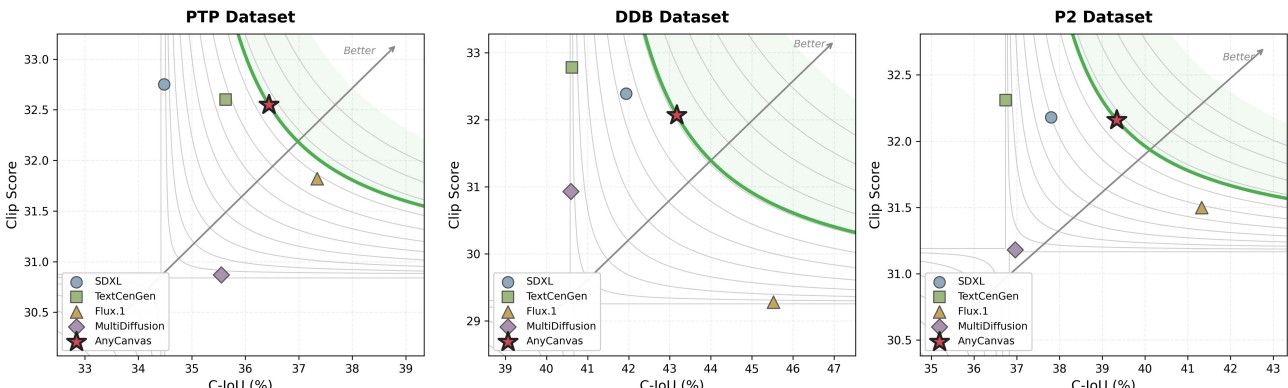

*Figure 5.* Trade-off analysis various datasets. The x-axis represents spatial accuracy (C-IoU) and the y-axis represents semantic quality (CLIP Score). Dashed contours indicate geometric mean iso-utility curves, which favor balanced performance over single-metric dominance. AnyCanvas (star) consistently resides on the highest utility curve, demonstrating a superior equilibrium compared to baselines that sacrifice one metric for the other.

min-max range of all evaluated methods:

$$\tilde{x} = \frac{x - x_{\min}}{x_{\max} - x_{\min}}, \quad \tilde{y} = \frac{y - y_{\min}}{y_{\max} - y_{\min}} \tag{27}$$

We then define the joint utility $U$ as the geometric mean of the normalized scores:

$$U(\tilde{x}, \tilde{y}) = \sqrt{\tilde{x} \cdot \tilde{y}} \tag{28}$$

**Justification.** The geometric mean is inherently sensitive to balance; it yields a high utility score only when *both* constituent metrics are substantial. This property makes it mathematically rigorous for identifying methods that achieve a robust trade-off, as represented by the convex iso-utility contours in Figure 5.

## C. Comparison with More Methods

We additionally compare three direct spatial control methods, with experimental results on different datasets shown below:

| Method | PTP | | | DDB | | |
|---|---|---|---|---|---|---|
| | C-IoU | CLIP Score | ICQS | C-IoU | CLIP Score | ICQS |
| FreeControl | 55.35 | 24.17 | 31.06 | 61.54 | 26.95 | 34.56 |
| LVD | 35.17 | 29.73 | 29.11 | 45.18 | 27.67 | 32.07 |
| BoxDiff | 35.49 | 31.70 | 31.10 | 39.40 | 31.84 | 33.44 |
| AnyCanvas (Ours) | 36.44 | 32.55 | 32.68 | 43.17 | 32.07 | 35.67 |

*Table 5.* Comparison with other spatial control methods on different datasets.

Specifically, similar to MultiDiffusion, FreeControl (Mo et al., 2024) relies on the assumption that the input contour matches the object's actual physical shape. For example, when generating a "Photo of a zebra walking in the field" within a heart-shaped mask, it simply fills the heart region with zebra-like stripes, which entirely disrupts the semantic structure. Consequently, its relatively high C-IoU comes at the cost of a significant drop in semantic fidelity (CLIP score falling to 24.17 and 26.95). Meanwhile, as bounding-box-based methods, LVD (Lian et al., 2024) and BoxDiff (Xie et al., 2023) perform adequately on near-rectangular canvases but degrade markedly on irregular shapes such as triangles, underperforming AnyCanvas across all metrics. In summary, these methods are not designed for the arbitrary canvas constraints addressed in this work, and their effectiveness under irregular shapes remains limited.

## D. Hyperparameter Analysis

We conduct a thorough empirical analysis of three key hyperparameters that govern the behavior of our potential field guidance: the smoothing factor $\sigma$ in (10), the global coefficient $\kappa$ in (16), and the **sensitivity parameter** $\lambda$ in (19). To isolate the impact of each parameter, we adopt a control-variable approach, varying one hyperparameter while keeping the others fixed at their default settings.

As illustrated in Figure 6, the quantitative results reveal distinct trends for each component. First, regarding the smoothing factor $\sigma$ (Figure 6a), performance metrics improve rapidly as the value increases, reaching a peak at approximately 30. This suggests that a moderate potential decay rate is essential for effective guidance. Second, the **global coefficient** $\kappa$ (Figure 6b) follows a similar trajectory. The scores rise and then plateau around $\kappa = 30$, indicating that while sufficient displacement strength is necessary, increasing it beyond this saturation point yields diminishing returns.

In contrast, the sensitivity parameter $\lambda$ (Figure 6c) exhibits a more volatile behavior. The performance peaks at $\lambda \approx 3$, achieving the optimal balance between spatial alignment and semantic integrity. However, as $\lambda$ increases beyond this threshold, we observe a noticeable deterioration: ICQS converges to a lower level, accompanied by a decline in both Canvas IoU and CLIP Score. This indicates that an overly aggressive scaling response (high $\lambda$) may excessively contract the attention maps, distorting the semantic structure. Based on these analyses, we set $\sigma = 30$, $\kappa = 30$, and $\lambda = 3$ as our default configuration.

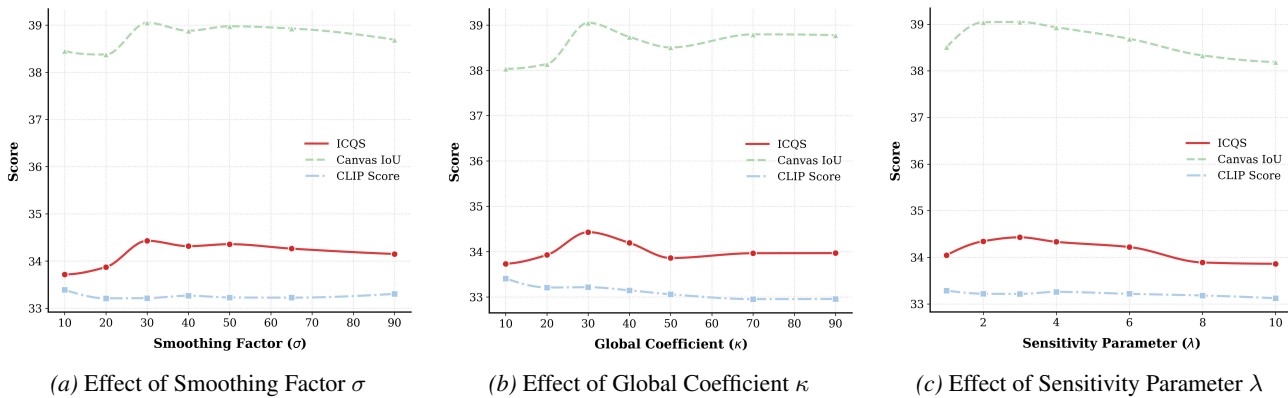

*(a)* Effect of Smoothing Factor $\sigma$      *(b)* Effect of Global Coefficient $\kappa$      *(c)* Effect of Sensitivity Parameter $\lambda$

*Figure 6.* Hyperparameter sensitivity analysis. We report the performance curves of ICQS, Canvas IoU, and CLIP Score by varying (a) the smoothing factor $\sigma$, (b) the global coefficient $\kappa$, and (c) the sensitivity parameter $\lambda$. The results indicate that the model achieves optimal performance with $\sigma = 30$, $\kappa = 30$, and $\lambda = 3$.

## E. Limitation and Failure Case Analysis

To illustrate the boundary conditions of our method, we provide a representative failure case in Figure 7. For instance, under extremely fragmented conditions like a checkerboard mask, the discontinuous nature of the input causes fluctuations in the potential field. This hinders the convergence of the Potential Field Guidance, making it challenging to satisfy highly scattered constraints while preserving structural integrity.

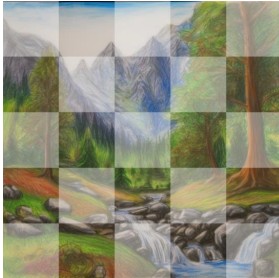

*Figure 7.* Case study.

