# OpenReview forum: "AnyCanvas: Potential Field Guidance for Training-Free Spatial Control in Text-to-Image Diffusion"
_ICML.cc/2026/Conference — ICML 2026 regular_

### Official Review · Reviewer_WQtN · 2026-03-10

**Soundness:** 2
**Presentation:** 3
**Significance:** 2
**Originality:** 2
**Overall Recommendation:** 4
**Confidence:** 4

**Summary:**

This paper proposes a training-free method for generating images that conform to arbitrary canvas shapes in text-to-image diffusion models. The approach converts a user-provided binary mask into a differentiable potential field derived from a signed distance function. The gradient of this potential field is used to guide the diffusion process by manipulating cross-attention maps through translation and scaling operations, encouraging semantic content to concentrate within the target region. The paper evaluates the method on several benchmarks and reports improvements in spatial adherence while maintaining comparable semantic alignment.

**Compliance With Llm Reviewing Policy:**

Affirmed.

**Final Justification:**

The rebuttal addressed most of concerns.

**Key Questions For Authors:**

See above.

**Limitations:**

Yes

**Strengths And Weaknesses:**

**Strength**

1. The writing is smooth and easy to follow.

2. The idea of converting a binary mask into a signed distance field and subsequently a potential field for spatial guidance is intuitive and elegant.

3. The approach is training-free and introduces only a small computational overhead according to the reported analysis.

**Weakness**

1. The performance gain appears relatively modest. For example, the improvements reported in Table 1 are small compared to the baselines. It is also unclear how many images are generated for each method when computing the reported metrics. The statistical significance of the improvements is therefore unclear.

2. The method is heuristic. It relies on several key hyperparameters, such as the smoothing factor, global coefficient, and sensitivity parameter.  Several other components like exponential displacement function, clamp scaling, token filtering threshold, and padding trick are engineering heuristics, not theoretically justified.

3. The paper claims that the proposed approach exhibits robust generalizability across different model backbones, but it only evaluates variants within the Stable Diffusion family. No evidence is provided to demonstrate it could work on top of other architectures.

---

> ### Author Rebuttal · Authors · 2026-03-30
>
> We sincerely thank you for your insightful and constructive comments, which have helped us improve the paper. For your concerns, we address them below:
>
> ## Weakness 1：
>
> In fact, our results demonstrate strong overall competitiveness. From an application perspective, seemingly small quantitative differences can still translate into a noticeable **"usability gap"** in practice. Some methods tend to overly optimize spatial control metrics: although achieving higher scores, this may come at the expense of semantic consistency.
>
> For example, in FreeControl (as noted by reviewer Savd), the generated results strictly stay within the mask but deviate significantly from the prompt, indicating that a single metric can be misleading and may yield inflated scores while lacking practical value (see Figure 1 at https://anonymous.4open.science/r/001anonymousfig-D629/Reviewer_WQtN.md). Similarly, for the first column in Figure 3, the CIoU(spatial adherence metric) difference between FLUX.1 (Figure a) and AnyCanvas (Figure b) is only 0.60. However, visually, the horse in Figure a exceeds the mask boundary, whereas Figure b better respects the structural constraint, showing that a single quantitative metric cannot fully capture structural adherence.
>
> Regarding **data scale**, we generated 3,096 images on PTP, 8,991 on DDB, and 14,688 on P2, totaling 26,775 images per method. This is competitive with prior work and ensures statistical significance and stability, further supporting the reliability and practical significance of our improvements. We will clarify this in the revised manuscript.
> ## Weakness 2：
>
> We thank the reviewer for the question and would like to clarify that the designs and hyperparameters in AnyCanvas are grounded in a physically motivated and robustness-oriented framework, rather than arbitrary or purely heuristic engineering choices.
>
> **The key hyperparameters** exhibit robustness across different conditions. As shown in Figure 6, the system maintains stable performance across a wide parameter range, and we used the same default settings (σ=30, κ=30, λ=3) across all datasets, masks and prompts in the main experiment. This demonstrates our hyperparameter settings are not scenario-specific heuristic tuning, but stable configurations aligned with the framework’s physical design principles.
>
> **The core components** can be viewed as physical boundary conditions and optimization constraints in mapping a continuous dynamical system to cross-attention guidance. The padding method corresponds to Dirichlet boundary conditions and an infinite potential barrier to prevent attention particle escape; clamp scaling corresponds to force softening to avoid distribution singularities and spatial collapse; the exponential displacement function simulates the Born–Mayer nonlinear repulsive model and soft barrier penalties in constrained optimization; and the token filtering threshold corresponds to the active set method for targeted constraint intervention. We will further incorporate these theoretical mappings in the revised manuscript to improve rigor.
>
> ## Weakness 3：
> We thank the reviewer for the constructive question regarding the generalizability of AnyCanvas. As a plug-and-play method, AnyCanvas inherently possesses strong generalization capabilities. To demonstrate that it is not limited to the Stable Diffusion (U-Net) family, we deployed it on mainstream pure Diffusion Transformer architectures, taking PixArt-$\alpha$ as an example. The experimental results are shown in the table below:
>
> | Dataset | Method | C-IoU | CLIP Score | ICQS |
> | :--- | :--- | :--- | :--- | :--- |
> | PTP | DiT | 35.06 | 32.77 | 31.39 |
> | PTP | **DiT + AnyCanvas** | 36.00 | 31.93 | **32.69** |
> | DDB | DiT | 33.25 | 30.09 | 28.00 |
> | DDB | **DiT + AnyCanvas** | 35.23 | 29.34 | **29.38** |
>
> As shown, compared with the DiT baseline, AnyCanvas achieves improvements in spatial adherence (C-IoU) while maintaining strong semantic fidelity (CLIP), leading to a gain in ICQS.(see cases in https://anonymous.4open.science/r/001anonymousfig-D629/Reviewer_WQtN.md Figure 2)  This provides strong empirical evidence that our potential field guidance mechanism can bridge architectural differences and achieve excellent spatial control on standard DiT models. We will incorporate relevant experimental results into the revised manuscript to further demonstrate the generalizability of AnyCanvas.
>
> We hope our responses can adequately address your concerns and earn your recoginition. We wish you all the best in your work and life.

---

> > ### Author Rebuttal · Reviewer_WQtN · 2026-04-03
> >
> > Thanks for the rebuttal. I raise score to 4.

---

> > > ### Author Response · Authors · 2026-04-06
> > >
> > > We are glad that our rebuttal has addressed your concerns. We will incorporate the constructive suggestions and the corresponding experiments into the revised manuscript. Thank you very much for your time and valuable feedback.

---

### Official Review · Reviewer_Feb2 · 2026-03-11

**Soundness:** 3
**Presentation:** 3
**Significance:** 3
**Originality:** 3
**Overall Recommendation:** 4
**Confidence:** 4

**Summary:**

This paper proposes AnyCanvas, a training-free spatial control framework for text-to-image generation under arbitary-shaped canvas. The core idea is to convert a binary bask into a potential field, and use this field to dynamically adjust the cross attention maps. Extensive experiments demonstrate the effectiveness of the proposed method.

**Compliance With Llm Reviewing Policy:**

Affirmed.

**Final Justification:**

The rebuttal has resolved my concerns.

**Key Questions For Authors:**

See weakness.

**Limitations:**

Further adaptation to modern architectures such as MMDiT.

**Strengths And Weaknesses:**

Strength:
1. The idea is novel and conceptually simple.
2. The writing of this paper is easy to follow.
3. Experimental results are very thorough and convincing. The author provides extensive results and ablations, as well as additional task analysis. The effectiveness of AnyCanvas is properly proved and demonstrated.

Weakness:
1. The comparisons across baselines seem not to be fair. Several baselines are not designed for arbitrary-shape control. Can the author compare it with other spatial-control methods, rather than only with general models such as FLUX.1 with prompt control?
2. Can AnyCanvas be applied to mainstream T2I models without cross-attention control? Modern models such as FLUX.1 use MMDiT for semantic and spatial attention. Applying to a widerly use architecture can further improve the soundness of AnyCanvas.

---

> ### Author Rebuttal · Authors · 2026-03-30
>
> We sincerely thank you for your insightful and constructive comments, which have helped us improve the paper. For your concerns, we address them below:
>
> ## Weakness 1：
>
> As described in Section 1, existing studies have **not** proposed dedicated solutions for this problem. Current spatial control methods can be divided into two categories. The first relies on **precise object contours** to constrain the generation region, such as **MultiDiffusion**（our baseline） and **FreeControl**（additional baseline）. However, these methods depend heavily on the consistency between the input contour and the target semantics; using only a canvas boundary instead of a true object contour often leads to **distortion and artifacts**.
>
> The second category controls object placement using **bounding boxes**, such as **TextCenGen** (our baseline), **LVD** (additional baseline), and **BoxDiff** (additional baseline). While effective for coarse localization, these methods struggle to handle irregular shapes or single-direction constraints (e.g., only inward or outward control), resulting in **limited flexibility**. As a result, they often require approximating irregular shapes with rectangular regions or discarding methods that are not applicable to the task.Despite this, we selected three additional spatial control methods as baselines.
>
> | Method | PTP | | | | | DDB | | |
> | :--- | :--- | :--- | :--- | :--- | :--- | :--- | :--- | :--- |
> | | C-IoU | CLIP Score | **ICQS** | | | C-IoU | CLIP Score | **ICQS** |
> | FreeControl | 55.35 | 24.17 | 31.06 | | | 61.54 | 26.95 | 34.56 |
> | LVD | 35.17 | 29.73 | 29.11 | | | 45.18 | 27.67 | 32.07 |
> | BoxDiff | 35.49 | 31.70 | 31.10 | | | 39.40 | 31.84 | 33.44 |
> | **AnyCanvas (Ours)** | 36.44 | 32.55 | **32.68** | | | 43.17 | 32.07 | **35.67** |
>
>
> Although these methods perform well on their respective tasks, they fail to jointly achieve strong spatial control and semantic alignment in arbitrary canvas generation, and thus still lag behind AnyCanvas in terms of ICQS.
>
>
>
> ## Weakness 2：
>
> We thank the reviewer for the constructive question regarding the generalizability of AnyCanvas. As a plug-and-play method, AnyCanvas inherently possesses strong generalization capabilities. To demonstrate that it is not limited to the Stable Diffusion (U-Net) family, we deployed it on a mainstream pure Diffusion Transformer(DiT) architecture (specifically PixArt-$\alpha$). The experimental results are shown in the table below:
>
> | Method | C-IoU | CLIP Score | ICQS |
> | :--- | :--- | :--- | :--- |
> | DiT | 35.06 | 32.77 | 31.39 |
> | **DiT + AnyCanvas** | 36.00 | 31.93 | **32.69** |
>
> As shown, compared with the DiT baseline, AnyCanvas achieves substantial improvements in spatial adherence (C-IoU) while maintaining strong semantic fidelity (CLIP), leading to a 4.14% gain in ICQS and demonstrating that our potential field guidance effectively **enables strong spatial control on DiT models despite architectural differences**. Regarding the adaptation to MMDiT architecture: Modern models adopt a multimodal transformer with joint attention, where visual and text tokens are processed within a unified attention mechanism. Although different from the explicit cross-attention mechanism in U-Net-based diffusion models, the joint attention matrix can be viewed as comprising Text–Text (TT), Text–Image (TI), Image–Text (IT), and Image–Image (II) interactions, with the TI component playing a key role in semantic-to-spatial alignment. Recent work (e.g., Rare-to-Frequent, ICLR 2025)[4] shows TI interactions are analogous to cross-attention in guiding spatial correspondence. Therefore, our method can be extended by operating on TI-related attention, where we extract attention maps and apply the potential field transformation before softmax to achieve spatial control. Further analysis and experiments will be included to validate generalization across architectures.
>
> We hope our responses can adequately address your concerns and earn your recoginition. We wish you all the best in your work and life.
>
> [1] Freecontrol: Training-free spatial control of any text-to-image diffusion model with any condition. CVPR 2024
>
> [2] LLM-grounded Video Diffusion Models. ICLR 2024
>
> [3] BoxDiff: Text-to-Image Synthesis with Training-Free Box-Constrained Diffusion. ICCV 2023
>
> [4]Rare-to-Frequent: Unlocking Compositional Generation Power of Diffusion Models on Rare Concepts with LLM Guidance.ICLR 2025 Spotlight

---

> > ### Author Rebuttal · Reviewer_Feb2 · 2026-04-03
> >
> > My concerns are fully addressed, and I maintain my positive score.

---

> > > ### Author Response · Authors · 2026-04-07
> > >
> > > We are pleased that our rebuttal has addressed your concerns. We will incorporate the relevant experiments into the revised manuscript. Thank you for your time and valuable feedback.

---

### Official Review · Reviewer_Savd · 2026-03-11

**Soundness:** 2
**Presentation:** 3
**Significance:** 3
**Originality:** 3
**Overall Recommendation:** 4
**Confidence:** 4

**Summary:**

The paper addresses the challenge of achieving precise, training-free spatial control in text-to-image diffusion generation, especially for irregular and complex-shaped canvases. To address this, this paper proceeds to present AnyCanvas, a plug-and-play framework based on a physics-inspired "Mask-to-Potential Field" paradigm. It converts binary masks into differentiable potential fields, using their gradients to dynamically adjust cross-attention maps via affine transformations during diffusion sampling. Extensive experiments confirm that AnyCanvas achieves a superior trade-off between spatial adherence and semantic fidelity.

**Compliance With Llm Reviewing Policy:**

Affirmed.

**Final Justification:**

The author's rebuttal addresses all my concerns, so I raise my score to weak accept.

**Key Questions For Authors:**

My final rating will depend on how well the authors address Weaknesses 1-3; a convincing rebuttal would likely lead me to lean towards a positive recommendation.

**Limitations:**

The authors can provide a failure case analysis to further clarity the boundary conditions of the proposed AnyCanvas and offer insights into its current limitations.

**Strengths And Weaknesses:**

### Strengths
1. The "mask-to-potential field" paradigm is an efficient, modular approach that does not require additional retraining, directly accommodating diverse diffusion-based generation models.
2. Multiple baselines (PTP, DiffusionDB, PartiPrompts) are employed to evaluate the proposed AnyCanvas.

### Weaknesses
1. Some baselines (e.g., FLUX.1 and TextCenGen) are adapted for spatial control using prompt engineering or partial module disabling, possibly limiting their optimal performance potential and making the comparison less rigorous.
2. This paper evaluates AnyCanvas against general-purpose models (i.e., SDXL, FLUX.1) and methods not inherently designed for arbitrary mask control (i.e., TextCenGen, MultiDiffusion). They may have overlooked the direct comparison: training-free layout control methods like FreeControl[a]、LMD[b] or BoxDiff[c] that support arbitrary shapes/masks.
3. The details for extracting semantic saliency maps to compute Canvas-IoU is not clear, which may influence metric reliability.

[a] Freecontrol: Training-free spatial control of any text-to-image diffusion model with any condition. CVPR 2024

[b] LLM-grounded Video Diffusion Models. ICLR 2024

[c] BoxDiff: Text-to-Image Synthesis with Training-Free Box-Constrained Diffusion. ICCV 2023

---

> ### Author Rebuttal · Authors · 2026-03-30
>
> We sincerely thank you for your insightful and constructive comments, which have helped us improve the paper. For your concerns, we address them below:
>
> ## Weakness 1：
>
> The problem setting of this paper is to *generate complete images that conform to arbitrary-shaped canvas constraints*. **No dedicated algorithmic** designs have been proposed for this problem in existing research. Therefore, although many methods perform well in their respective domains, they are not well-suited to our task and often require adaptation.
>
> For example, **TextCenGen** guides generation outside a specified region, but its constraint is limited to a **rectangle** with a fixed direction, restricting its use in scenarios such as product packaging or logo design; thus, it requires adaptation as a baseline. We also use **FLUX.1** to show that relying **solely on text prompts** cannot effectively handle arbitrary canvases. Meanwhile, methods like **MultiDiffusion**, although capable of handling irregular inputs, depend on manual guidance and input–shape alignment, and may still produce **low-quality** results and artifacts when only a canvas mask is provided.
>
> In summary, although some methods appear applicable, their effectiveness on this task remains limited.
>
> ## Weakness 2：
>
> We have evaluated the three direct spatial control methods as suggested, with the experimental results on different datasets shown below:
>
> | Method | PTP | | | | | DDB | | |
> | :--- | :--- | :--- | :--- | :--- | :--- | :--- | :--- | :--- |
> | | C-IoU | CLIP Score | **ICQS** | | | C-IoU | CLIP Score | **ICQS** |
> | FreeControl | 55.35 | 24.17 | 31.06 | | | 61.54 | 26.95 | 34.56 |
> | LMD | 35.17 | 29.73 | 29.11 | | | 45.18 | 27.67 | 32.07 |
> | BoxDiff | 35.49 | 31.70 | 31.10 | | | 39.40 | 31.84 | 33.44 |
> | **AnyCanvas (Ours)** | 36.44 | 32.55 | **32.68** | | | 43.17 | 32.07 | **35.67** |
>
> Specifically, similar to MultiDiffusion, **FreeControl** relies heavily on the assumption that the input contour matches the object's actual physical shape. For example, when generating a "Photo of a zebra walking in the field" within a heart-shaped mask, it merely fills the heart with zebra stripes, which completely destroys the semantic structure(see https://anonymous.4open.science/r/001anonymousfig-D629/Reviewer_Savd.md Figure.1). Thus, its artificially high C-IoU comes at the cost of a catastrophic drop in semantic fidelity (CLIP drops to  24.17 and 26.95).  Furthermore, as **BoxDiff** and **LMD** are bounding-box-based methods, like TextCenGen, it performs well on near-rectangular canvases but deteriorates significantly on irregular shapes (e.g., triangles), falling short of AnyCanvas across all metrics. In conclusion, as these methods are not designed for the problem scenario of Anycanvas, their effectiveness under arbitrary and irregular canvas constraints remains poor.
>
> ## Weakness 3：
>
> The saliency detection method adopted in our evaluation is based on BASNet [1] as cited in our paper. We will include the underlying principles of BASNet in the revised version. Specifically, the underlying principle of BASNet relies on a predict-refine architecture, where a residual refinement module enhances the coarse saliency map $S_{coarse}$ by learning the residual $S_{residual}$ to output an accurate map: $S_{refined}=S_{coarse}+S_{residual}$. Furthermore, its optimization is governed by a hybrid loss: $l = l_{bce} + l_{ssim} + l_{iou}$. This specific combination ensures pixel-wise convergence ($l_{bce}$), enforces sharp and accurate structural boundaries ($l_{ssim}$), and maintains consistent foreground region predictions ($l_{iou}$). Thus, it provides highly reliable spatial constraints for our evaluation.
>
> Existing related works (e.g., Desigen [2], TextCenGen [3]) have also employed BASNet for saliency map extraction; therefore, we follow this common practice to maintain consistency with prior studies.
>
> ## Limitation：
>
> We thank the reviewer for the constructive suggestion. We conduct a case study to analyze the failure modes of our method. For example, the fragmented and discontinuous nature of a checkerboard mask causes fluctuations in the potential field, hindering the convergence of the Potential Field Guidance and making it difficult to satisfy scattered constraints while preserving structural integrity(see https://anonymous.4open.science/r/001anonymousfig-D629/Reviewer_Savd.md Figure2). This highlights the boundary conditions and limitations of the current method. We have added this failure case analysis to the revised manuscript to better clarify the applicability of AnyCanvas.
>
> We hope our responses can adequately address your concerns and earn your recoginition. We wish you all the best in your work and life.
>
> [1] Boundary aware salient object detection. [IEEE/CVF 2019].
>
> [2] Desigen: A Pipeline for Controllable Design Template Generation. ICLR2024
>
> [3]TextCenGen: Attention-Guided Text-Centric Background Adaptation for Text-to-Image Generation. ICML2025

---

> > ### Author Rebuttal · Reviewer_Savd · 2026-04-02
> >
> > I appreciate the additional experiments, but the rebuttal does not fully resolve my concerns regarding the rigor of the baseline setup in this new problem setting. Furthermore, given that the performance gains over the specified baselines in the rebuttal are relatively marginal, I choose to maintain my original score.

---

> > > ### Author Response · Authors · 2026-04-02
> > >
> > > Thank you for your continued engagement and feedback. We are glad that you appreciate the additional experiments. Regarding your follow-up questions on the rigor of the baseline comparisons and the magnitude of the quantitative improvements, we provide a clear clarification here:
> > >
> > > **1. Rigorous Baseline Setup**
> > >
> > > To conduct a stricter evaluation under the new problem setting of "generate complete images that conform to arbitrary-shaped canvas constraints," we have adopted your suggestion and formally incorporated the three layout control methods (FreeControl, LMD, and BoxDiff)as core comparison baselines in the revised manuscript. By introducing these directly relevant methods, we have established a more systematic and rigorous evaluation framework.
> > >
> > > **2. "The Usability Gap" of Baselines**
> > >
> > > Under this stricter evaluation baselines, the differences in some quantitative metrics appear limited, but the baselines actually reflect a significant "usability gap." Relying solely on spatial metrics can be somewhat misleading, as existing methods primarily exhibit two typical failure modes:
> > >
> > > * **Mode 1: Boundary Overflow (Visually unusable).** Bounding-box-based methods (such as LMD, BoxDiff, etc.) are constrained by rectangular priors. In certain generation cases (see https://anonymous.4open.science/r/002-3E86/Reviewer_Savd.md Figure1), although the absolute area of the out-of-bounds region is small, making the difference in C-IoU(spatial constraint metric) between the baseline and ours less than 1%, key structures (e.g., the giraffe's head) clearly exceed the irregular boundary, which is unacceptable from a practical application perspective. In contrast, AnyCanvas's irregular potential field constraints can more accurately confine the subject within the target structure.
> > > * **Mode 2: Semantic Degeneration (Structural distortion).** Although FreeControl can strictly adhere to boundaries and achieve higher C-IoU scores(spatial constraint metric), it often does so at the expense of semantic structure, exhibiting a degeneration phenomenon akin to "texture filling." For example, in the case of generating a "Photo of a zebra walking in the field" within a heart-shaped mask (see https://anonymous.4open.science/r/002-3E86/Reviewer_Savd.md Figure2), it fails to generate a complete zebra structure and merely fills the heart-shaped region with "zebra textures." Such high spatial scores built upon the absence of structure lack practical application value. Furthermore, we have provided additional similar failure cases (where despite ostensibly satisfying the mask constraint, the semantic meaning visibly deviates. See https://anonymous.4open.science/r/002-3E86/Reviewer_Savd.md Figure3) to further illustrate the prevalence of this issue. In comparison, AnyCanvas's improvement is not a simple numerical gain, but a substantial enhancement in generation quality.
> > >
> > > **3. Underlying Mechanism for our Method's Advantages**
> > >
> > > AnyCanvas bridges this gap through its fundamental mechanism. First, by constructing a potential field tailored for irregular masks, it prevents the boundary overflow of Mode 1. Second, unlike existing methods, AnyCanvas treats the semantic attention map as a unified entity. Guided by the "Mask-to-Potential Field," we modulate the cross-attention activations using a continuous affine transformation (integrating force-driven translation and isotropic scaling) via inverse coordinate mapping. This geometric operation mathematically guarantees that the relative spatial relationships and internal topological structure of the subject's features are strictly preserved. Consequently, AnyCanvas fundamentally avoids the structural tearing of Mode 2, making it the only method among the comparisons capable of adapting a complete subject into complex shapes.
> > >
> > > In summary, the advantages of AnyCanvas stem not only from numerical gains, but also from truly achieving structurally complete and precise spatial control while fundamentally guaranteeing semantic quality.We will comprehensively supplement these analyses in the final version.

---

### Official Review · Reviewer_Anm3 · 2026-03-13

**Soundness:** 3
**Presentation:** 3
**Significance:** 3
**Originality:** 4
**Overall Recommendation:** 4
**Confidence:** 3

**Summary:**

The paper proposes AnyCanvas to enable spatial control in T2I diffusion without tuning. It maps binary masks to a differentiable potential field via SDF and guides attention maps using gradient-based affine transformations.

**Compliance With Llm Reviewing Policy:**

Affirmed.

**Final Justification:**

I appreciate the authors’ detailed response, which addressed my specific technical queries. However, I have decided to maintain my score of 4. While the proposed method is technically sound and the paper is worth accepting, I still have lingering reservations about its broader generalizability and the limited practical impact of the marginal quantitative gains. Overall, it is a solid contribution, but a weak accept accurately reflects my current assessment.

**Key Questions For Authors:**

please refer to the weakness

**Limitations:**

The provided examples predominantly feature bounding boxes placed in the center of the canvas. This prevents a proper assessment of the method’s robustness against the known “center bias” of diffusion models. It remains unclear if the method can effectively guide attention to generation targets when they are positioned near the image boundaries or corners.

**Strengths And Weaknesses:**

Strengths
1. Modeling spatial constraints as a continuous potential field is theoretically cleaner than discrete masking or token weighting. It provides a smooth landscape for backward guidance.
2. Being training-free is significant. The compatibility across varied SD architectures verifies robustness.

Weakness
1. The method treats the attention map as a single particle, which likely fails when the prompt describes a group (e.g., “a flock of birds”) or interacting objects (“a cat and a dog”). Reducing multiple entities to a single centroid seems too reductive and may cause layout collapse.
2. I suspect the potential field logic struggles with shapes containing holes (e.g., a donut shape). The “center of mass” guidance might drag the generation right into the hole, which contradicts the mask’s intent.
3. The quantitative improvements over baselines look rather incremental. Is the added complexity of calculating SDFs and gradients really justified given the limited boost in metric performance?

---

> ### Author Rebuttal · Authors · 2026-03-30
>
> We sincerely thank you for your insightful and constructive comments, which have helped us improve the paper. For your concerns, we address them below:
>
> ## Weakness 1：
>
> In summary, our method can effectively handle such generation scenarios（see examples https://anonymous.4open.science/r/001anonymousfig-D629/Reviewer_Anm3.md Figure.1）. In our design, AnyCanvas inherently operates as a **multi-particle dynamic system**. As described in the **Selective Attention Modulation** of Section 4.3, the potential field guidance acts on individual semantic units rather than the global map. For example, when generating “a cat and a dog,” the system extracts centroids for the feature clusters of “cat” and “dog,” treating them as two independent particles within the potential field, each seeking its own low-potential region without interference.
>
> For collective entities such as “a flock of birds,” although it is difficult to decompose them into individual elements at the token level, our **Conflict Detection** mechanism remains effective: when the overall feature distribution exceeds the mask boundaries, potential field guidance is applied to pull the group back within the mask. Therefore, AnyCanvas can still handle generation tasks involving collective concepts within arbitrary canvases.
>
> ## Weakness 2：
>
> For your concern that the generation might indeed be "dragged" into the hole of a donut shape mask, we argue the potential field design in AnyCanvas **can inherently avoids this risk**.
> As detailed in our **Mask-to-Potential Field** construction (Section 4.1), for a donut shape mask with a hollow topology, the "hole" in the center belongs to the external region of the mask ($M=0$), which is explicitly mapped as a high potential barrier. This creates a powerful **repulsive field**. Consequently, when the predicted trajectory of attention particles approaches this hollow area, they experience a repulsive force derived from the potential gradient. This force "pushes" the particles away from the high-potential hole and into the annular low-potential region (where $M=1$), rather than pulling them into the center.
> To quantitatively validate this, we conducted targeted tests on a sub-dataset specifically consisting of these challenging donut shape mask, with results summarized below:
> | Method | C-IoU | CLIP Score | ICQS |
> | :--- | :--- | :--- | :--- |
> | Baseline | 27.97 | 32.99 | 26.81 |
> | **AnyCanvas (Ours)** | 29.43 | 32.63 | **27.51** |
>
> The experimental data demonstrate that AnyCanvas maintains the semantic information even when dealing with such complex topological shapes（see cases https://anonymous.4open.science/r/001anonymousfig-D629/Reviewer_Anm3.md Figure.2）.
>
> ## Weakness 3：
>
> In fact, our results demonstrate strong overall competitiveness. From an application perspective, small quantitative differences can still correspond to **significant gaps in visual usability**, indicating a potential **“usability gap”** in practice. Some methods overly optimize spatial control metrics; although achieving higher scores, this may come at the cost of semantic consistency. For example, in the FreeControl method mentioned by reviewer Savd, the generated results strictly stay within the mask but deviate significantly from the prompt. In such cases, a single metric may be misleading due to inflated scores, while the results **lack practical value** （see qualitative examples https://anonymous.4open.science/r/001anonymousfig-D629/Reviewer_Anm3.md Figure.3）
> Similarly, for the prompt “Photo of a horse sitting in the desert” with a house-shaped mask in Figure 3, the  CIoU(spatial adherence metric)  difference between FLUX.1 (Figure.a) and AnyCanvas (Figure.b) is only 0.60. However, visually, the horse's head in Figure.a exceeds the mask boundary, whereas Figure.b better constrains the subject within the target region. This shows that **a single quantitative metric cannot fully capture structural constraint satisfaction**. Therefore, both quantitative and qualitative evaluations are necessary. Their combined analysis shows that AnyCanvas achieves consistently superior performance and the strongest overall competitiveness.
>
> ## Limitation：
>
> We thank the reviewer for the insightful concern. In most practical scenarios, the key requirement is to generate content within a given shape, while its **precise placement is often not critical**. Therefore, placing the mask at the center is a natural and convenient choice.
>
> Nevertheless, our method is **not limited** to centered layouts. As shown in Figure 4 (third and fourth columns) in paper, we provide off-center examples, demonstrating that the proposed potential field guidance can effectively handle such cases. These results indicate that AnyCanvas can guide generation beyond central regions when needed.
>
> We hope our responses can adequately address your concerns and earn your recoginition. We wish you all the best in your work and life.

---

> > ### Author Rebuttal · Reviewer_Anm3 · 2026-04-03
> >
> > I would maintain my score, thanks.

---

> > > ### Author Response · Authors · 2026-04-07
> > >
> > > We are delighted that our rebuttal could address your concerns. Thank you very much for your time and efforts.

---

### Decision · Program_Chairs · 2026-04-30

**Decision:**

Accept (regular)

**Comment:**

The paper introduces AnyCanvas, a training-free framework that leverages a Mask-to-Potential Field paradigm to transform binary masks into a differentiable potential field, enabling generation to naturally converge within target regions of arbitary geometry. Reviewers Feb2 and WQtN recognized the novelty of the work and the quality of the writing. Reviewers Anm3, Savd, and WQtN also acknowledged the method’s flexibility and its training-free nature. However, several concerns were also raised. Reviewer Anm3 questioned the feasibility of extending the proposed method to more complex shapes. Reviewers Savd and Feb2 noted that some comparisons were not entirely fair, as the compared baselines were designed for general-purpose generation rather than other training-free layout control methods. Reviewers Feb2 and WQtN also have concern if the proposed method could be generalized to other modern diffusion architectures. In response, the authors provided additional experimental results, including evaluations on more specialized baselines, more modern diffusion architectures, experiments on the suggested scenarios, and a detailed failure-case analysis. After the rebuttal, the reviewers’ concerns were well-addressed, and the paper finally received four unanimous weak accept recommendations after the reviewer discussion. Considering the overall novelty, technical merit, and the authors’ thorough response to the reviewers’ concerns, the paper is recommended for acceptance.